# Identification of Novel Circular RNAs of the Human Protein Arginine Methyltransferase 1 (*PRMT1*) Gene, Expressed in Breast Cancer Cells

**DOI:** 10.3390/genes13071133

**Published:** 2022-06-24

**Authors:** Maria Papatsirou, Marios A. Diamantopoulos, Katerina Katsaraki, Dimitris Kletsas, Christos K. Kontos, Andreas Scorilas

**Affiliations:** 1Department of Biochemistry and Molecular Biology, Faculty of Biology, National and Kapodistrian University of Athens, 15701 Athens, Greece; papatsir@biol.uoa.gr (M.P.); imdiamantop@biol.uoa.gr (M.A.D.); kkatsaraki@biol.uoa.gr (K.K.); ascorilas@biol.uoa.gr (A.S.); 2Laboratory of Cell Proliferation & Ageing, Institute of Biosciences and Applications, National Centre for Scientific Research “Demokritos”, 15341 Aghia Paraskevi, Greece; dkletsas@bio.demokritos.gr

**Keywords:** circRNAs, PRMT1, sequencing, breast cancer, alternative splicing, back-splice junction, RNA-binding proteins, miRNAs, bioinformatics, RNA modifications

## Abstract

Circular RNAs (circRNAs) constitute a type of RNA formed through back-splicing. In breast cancer, circRNAs are implicated in tumor onset and progression. Although histone methylation by PRMT1 is largely involved in breast cancer development and metastasis, the effect of circular transcripts deriving from this gene has not been examined. In this study, total RNA was extracted from four breast cancer cell lines and reversely transcribed using random hexamer primers. Next, first- and second-round PCRs were performed using gene-specific divergent primers. Sanger sequencing followed for the determination of the sequence of each novel *PRMT1* circRNA. Lastly, bioinformatics analysis was conducted to predict the functions of the novel circRNAs. In total, nine novel circRNAs were identified, comprising both complete and truncated exons of the *PRMT1* gene. Interestingly, we demonstrated that the back-splice junctions consist of novel splice sites of the *PRMT1* exons. Moreover, the circRNA expression pattern differed among these four breast cancer cell lines. All the novel circRNAs are predicted to act as miRNA and/or protein sponges, while five circRNAs also possess an open reading frame. In summary, we described the complete sequence of nine novel circRNAs of the *PRMT1* gene, comprising distinct back-splice junctions and probably having different molecular properties.

## 1. Introduction

Protein activity is determined by many post-translational modifications that occur on amino acid residues such as arginine and lysine [1]. Protein methylation is a modification that is greatly implicated in the regulation of major cellular processes, and arginine methylation, in particular, is implicated in several biological activities, including transcriptional control, RNA metabolism, and DNA damage repair [2]. Protein arginine methyltransferase 1 (PRMT1) is the predominant methyltransferase in mammalian cells, as it accounts for up to 85% of total cellular PRMT activity [3]. The main substrate of the *PRMT1* gene products is histone H4 [4]. The human PRMT1 is highly conserved in eukaryotes and has three functional domains in its canonical structure [5]. The first domain is an N-terminal methyltransferase domain, the second is a C-terminal β-barrel domain, and the third domain develops an α-helical dimerization finger [6]. Scorilas et al. previously described the genomic organization and expression pattern of the *PRMT1* gene. It is located on chromosome 19 (19q13.3), and several events of alternative splicing lead to at least seven *PRMT1* splice variants [7].

PRMT1 is proven to contribute to the pathogenesis of many types of cancer and is predominantly characterized as an early catalyst of breast cancer [8]. In the Western world, breast carcinoma is the deadliest cancer in women, since only a small percentage of cases are discovered in time. Breast cancer susceptibility is influenced by several factors, including hereditary, endocrine, and environmental factors [9]. So far, 10 genes have been identified and linked to inherited breast cancer, all of which are part of a pathway that maintains genomic integrity [10]. Despite this, nearly half of all patients with familial breast cancer are not linked to any of these genes [10]. *PRMT1* gene expression is substantially higher in breast tumor samples than in healthy breast tissues, and several studies evidence its role in regulating the metastatic potential of breast cancer, as well as therapy efficacy [11,12].

The critical involvement of *PRMT1* in the tumorigenesis and progression of breast cancer, as well as the preservation of stem cell-like features in breast cancer cells, is considered a very intriguing and complex process [13,14]. For example, in triple-negative breast cancer cells, PRMT1-dependent EGFR methylation upregulates many signaling pathways, including those involving MAPK1, AΚΤ1, and STAT3 [13]. Similarly, PRMT1 methylation of BRCA1 impacts its capacity to interact with specific partners (e.g., STAT1), and as a result, its tumor-suppressive action is severely affected [14]. Moreover, a mechanism that activates ZEB1 is the asymmetric dimethylation of H4R3 by PRMT1 at the *ZEB1* promoter, which promotes migration, invasion, and the acquisition of stem cell properties of breast cancer cells [15]. Considering the variety of PRMT1 substrates and the intricate pathways in which it is involved, the existence of only a few alternative transcripts of the *PRMT1* gene is under question. Owing to the recent advances in sequencing technologies, a new type of circular transcripts has been in the spotlight and could very likely be responsible for several mechanisms of *PRMT1* action.

Circular RNAs (circRNAs) are produced through back-splicing, in which the splice donor site is located downstream to the splice acceptor site and comprise one or more exons and introns of a gene [16,17,18]. Scientists discovered that circRNAs primarily act endogenously as microRNA (miRNA) sponges, protein sponges and scaffolds, while they can also regulate their parental gene transcription and produce functional proteins [19,20]. Alternative splicing is often implicated in the production of several circRNAs from the same gene, and diverse splicing events are a major driver of differential circRNA mechanism of action [21]. A growing number of circRNAs have been identified and studied in recent years due to breakthroughs in sequencing protocols, and a plethora of circRNAs play a major role in the progression of human disorders, including Alzheimer’s disease, cardiovascular disease, diabetes, and cancer [22,23,24,25]. Especially in breast tumors, circRNAs are evidenced to modulate major cellular processes such as migration, invasion, and metastasis, and affect therapy receptivity and patients’ prognosis [20,26]. Moreover, over the past few years, numerous circRNAs from different genes have been identified in breast cancer, the proteins of which are important regulators in different cellular processes. Some indicative examples are circ-HER2/HER2-103, circCD44, and circACTN4 [27,28,29]. However, there have been no studies on circRNAs deriving from the *PRMT1* gene so far.

In this study, we combined customized PCR protocols and Sanger sequencing with extensive bioinformatics analysis to elucidate the expression pattern of *PRMT1* circRNAs in four breast cancer cell lines and untangle the unique alternative splicing events that occur during the back-splicing process. From this procedure, novel *PRMT1* circRNAs were identified, which exhibit differential expression profiles. Moreover, the characteristics of these novel circRNAs were investigated, as well as their mechanism of action; interestingly, all novel *PRMT1* circRNAs are predicted to interact with miRNAs and RBPs. These findings shed light on the transcriptional potential and alternative splicing pattern of *PRMT1*, and the identification of novel circRNAs from this gene could pave the way for advances in the field of breast cancer biomarkers and/or therapeutic targets.

## 2. Materials and Methods

### 2.1. Culture of Breast Cancer Cell Lines

Four elaborately characterized and well-studied in the context of transcriptomic analysis breast cancer cell lines were chosen in this study. The BT-20, MCF-7, MDA-MB-231, and MDA-MB-468 breast cancer cell lines, which are of distinct characteristics and molecular subtypes (Appendix A) were purchased from the American Type Culture Collection (ATCC^®^) and cultured in an incubator at 37 °C and adjusted CO_2_ concentration of 5%. The BT-20 and MCF-7 cell lines were cultured in DMEM Low Glucose supplied with glutamine [2 mM], penicillin/streptomycin (100 U/mL) and fetal bovine serum (10%). For the MCF-7 cell line, human recombinant insulin was added to the culture medium at a final concentration of 0.01 mg/mL. The cell lines MDA-MB-231 and MDA-MB-468 were cultured in DMEM High Glucose supplied with glutamine (2 mM), penicillin/streptomycin (100 U/mL), fetal bovine serum (10%), and non-essential amino acids (1X). All cell culture materials were purchased from Biowest (Nuaillé, France).

### 2.2. Total RNA Extraction and Reverse Transcription

Following sample homogenization, total RNA was isolated from each cell line using the TRItidy G™ Reagent (AppliChem GmbH, Darmstadt, Germany). All RNA samples were diluted in DEPC-treated H_2_O and stored at −80 °C until further use. The quantity and purity of all RNA samples were evaluated spectrophotometrically, using the BioSpec-nano Micro-volume UV-Vis Spectrophotometer (Shimadzu, Kyoto, Japan). More specifically, the absorbance at 260 nm was measured for the assessment of RNA concentration, and the absorbance at 280 nm and 230 nm were used for the estimation of protein and phenol contamination. The integrity of all RNA samples was assessed by agarose gel electrophoresis.

Then, reverse transcription was carried out, using 3 μg of each RNA sample and random hexamer primers (New England Biolabs Ltd., Hitchin, UK), which is a mixture of single-stranded random hexanucleotides with 5′- and 3′-hydroxyl ends. In this way, the cDNA produced from each sample was derived not only from mRNA molecules but from circRNAs as well. Each reaction mixture contained, besides the RNA volume, 1 μL of random hexamer primers, 1 μL 10 mM dNTP Mix (10 mM of each dATP, dGTP, dCTP and dTTP at neutral pH) and DEPC-treated H_2_O at a final volume of 10 μL. The mixture was heated at 65 °C for 5 min, and was quickly transferred on ice for 1 min. Next, in each RNA sample, 1 μL of 5X First-Strand Buffer, 0.1 M DTT, 40 units RNaseOUT™ recombinant RNase inhibitor (Invitrogen™, Thermo Fisher Scientific Inc., Waltham, MA, USA) and 200 units of Reverse Transcriptase were added, resulting in a final reaction volume of 20 μL. Finally, each reaction was inactivated at 70 °C for 15 min. The reaction was conducted using the M-MLV reverse transcriptase (Invitrogen™), according to the manufacturer’s instructions. The process was performed in a MiniAmp Thermal Cycler (Applied Biosystems™, Thermo Fisher Scientific Inc.), resulting in a cDNA volume of 20 μL. The reverse transcription assay’s efficiency was checked by amplifying the *GAPDH* main variant using the produced cDNAs as templates. Next, a breast cancer cDNA pool was created, by mixing equal volumes of each cDNA deriving from the 4 breast cancer cell lines.

### 2.3. Primer Design and Two-Round PCR Assays

To selectively amplify cDNAs that originate from circRNAs of the *PRMT1* gene, and not its linear RNA variants, divergent primers were designed (Appendix A), as previously described [30]. These primers are facing outward, in contrast to the conventionally used convergent primers, and they are specific for exons of the *PRMT1* gene. The principle of divergent primers design is shown in Appendix A. The cDNAs and the cDNA pool were then subjected to first-round PCR, and the PCR products of these reactions were diluted at a ratio of 1:100 in nuclease-free H_2_O, at a final volume of 100 μL. Then, nested and semi-nested PCR was conducted, using 1 μL of the diluted products as template and an inner, second pair of divergent primers for each exon. The purpose of the second-round PCR was to increase the specificity and sensitivity for the *PRMT1* gene, along with the PCR yield (Appendix A). The reaction mix contained 1 unit of KAPA Taq DNA Polymerase (KAPA Biosystems Inc., Woburn, MA, USA), KAPA Taq Buffer A, at a final MgCl_2_ concentration of 1.5 mM, 0.2 mM dNTPs, and divergent primers at a final concentration of 0.4 μM. All first and second-round PCR assays were conducted in a MiniAmp Thermal Cycler (Applied Biosystems™), under the following cycling conditions: a denaturation step at 95 °C for 3 min, followed by 35 cycles of 95 °C for 30 s, an annealing step at the optimal Ta for each primer pair for 30 s, 72 °C for 1 min, and a final elongation step at 72 °C for 1 min.

### 2.4. Agarose Gel Electrophoresis and Sanger Sequencing

All of the aforementioned nested and semi-nested PCR products were electrophoresed on 2% agarose gels in equal volumes (10 μL) and visualized under UV light after ethidium bromide staining. Afterward, bands were properly excised from the gel and purified using a Gel and PCR Clean-up kit (Macherey-Nagel GmbH & Co. KG, Duren, Germany) and spin columns, as per the manufacturer’s instructions. Briefly, for each 100 mg of agarose gel, 200 μL Buffer NTI was used to solubilize the gel slice. Next, the samples were centrifuged for 30 sec at 11,000× *g*, and two steps of wash with Buffer NT3 and centrifugation were performed. At the final step, 30 μL of Buffer NE were added and each sample was centrifuged for 1 min at 11,000× *g* in order to elute the PCR product. The concentration of the extracted and purified PCR products was evaluated using a Qubit 2.0 fluorometer (Invitrogen™). Each of the purified products was subjected to two-sided Sanger sequencing for amplicon sequence verification, using as initiator sequences the forward and reverse divergent primers of each nested or semi-nested PCR. The sequencing data were manually analyzed, by annotating the raw DNA sequences using the latest version of the *PRMT1* gene sequence and identifying the exonic regions. The identification of a unique circRNA was confirmed only if the back-splice junction was present in the resulting reads from Sanger sequencing. The full experimental procedure that was followed is depicted in Figure 1.

### 2.5. Bioinformatics Analysis for the Prediction of the Interactions of the Novel PRMT1 CircRNAs with MiRNAs and RBPs

To elucidate the function of the novel *PRMT1* circRNAs that were identified and clarify the mechanisms in which they might be implicated, various bioinformatics tools and algorithms were utilized. Firstly, the custom prediction tool of the miRDB database was used [31], in order to predict the binding sites for miRNAs that the novel circRNAs possess. The sequences of the circRNAs were submitted and a list of miRNAs that could bind to their sequence was provided. Only the miRNAs with the two highest probability scores for each circRNA were chosen for downstream analysis. Taking into consideration the results from miRDB, pathway analysis was conducted through the KEGG Pathway database [32], aiming to identify regulatory cascades in which the novel circRNAs might be implicated or regulate.

Next, the putative sites for RNA-binding proteins (RBPs) and their respective motifs were predicted with the RBPmap tool [33], where the input is the custom RNA sequence and the output is an extensive list of RBPs, the motif that they recognize, as well as the Z-score and *p*-value of the prediction. The Z-score measures the deviation of the site’s weighted rank (WR) score from the mean score calculated using the genome-specific background. The Z-score is coupled to a *p*-value which represents the probability of obtaining a specific Z-score considering a normal one-tailed distribution. The RBPmap tool requires that the final WR score of a site will be significantly greater (*p* < 0.050) than the mean score calculated for the background, in order to consider this site as a predicted binding site [33].

As a means of further verifying the predicted interactions between these RBPs and the *PRMT1* circRNAs, CLIP sequencing data were utilized from two public databases. Considering that all 9 novel *PRMT1* circRNAs are firstly identified in this study, there is no CLIP sequencing data that verify the interaction between the circRNAs and the RBPs that are predicted to bind to them. Alternatively, we searched for RBP-mRNA interactions, by selecting the *PRMT1* mRNA, as an indirect way of verifying the binding of these RBPs to *PRMT1* sequences. Data were queried in the CLIPdb database by searching the RBP name, and in the ENCORI database, through the “RBP-Target” module, by selecting the RBP of interest and *PRMT1* as the target [34,35].

### 2.6. Bioinformatics Analysis for the Elucidation of the Structure and Features of the Novel PRMT1 CircRNAs

In order to predict the secondary structures and base pair probabilities of the novel *PRMT1* circRNAs, the RNAfold server of the ViennaRNA Package was employed [36]. The parameters that were selected were the minimum free energy and partition function of the algorithm, the avoidance of isolated base pairs, as well as the “assume RNA molecule to be circular” feature of the advanced options. Then, the prediction of the N6-methyladenosine (m^6^A) modification sites on the *PRMT1* circRNA sequences followed, considering that this post-transcriptional methylation is mainly implicated in the regulation of circRNA translation in a 5′ cap-independent manner. For this purpose, the circRNA sequences were submitted in FASTA format in the SRAMP prediction server [37], and the analysis of RNA secondary structure was also chosen so that the m^6^A sites would be visualized.

Next, the ORF Finder bioinformatics tool was utilized for the query of open reading frames (ORFs) in the nucleotide sequences of the novel circRNA transcripts. Considering that the sequences of the circRNAs should be submitted as linear, we had to “cut” them in a stochastic manner. Each prediction was conducted twice, where the sequences were cleaved at the back-splice junction and at the adjacent splice site. ORFs were queried in all reading frames as long as they begin with the codon ATG, and only the ORFs that produce putative peptides of more than 50 amino acids were considered. This way, the amino acid sequences of the novel PRMT1 protein isoforms were deduced. Clustal Omega (1.2.4.) was utilized for the alignment of the amino acid sequences, which implements seeded guide trees and Hidden Markov Model profile–profile techniques [38]. Finally, based on these sequences, three-dimensional (3D) structure models were built by the I-TASSER server [39], using multiple threading alignments and structural simulations. An estimation of the accuracy of every prediction is based on the confidence score of the modeling. The 3D structure models with the highest prediction score that were generated by I-TASSER were visualized by the PyMOL molecular visualization system.

## 3. Results

### 3.1. Expression Analysis of PRMT1 CircRNAs in Breast Cancer Cell Lines and Identification of Nine Novel CircRNAs

Through the optimization of the PCR protocols that were implemented, qualitative expression analysis of the circRNAs that are produced from *PRMT1* was performed in four breast cancer cell lines of distinct molecular subtypes. In particular, we assessed the *PRMT1* circRNAs expression pattern in the MCF-7, MDA-MB-231, MDA-MB-468, and BT-20 cell lines, and discovered that a wide range of diverse circRNAs are produced from *PRMT1*, and each cell line is characterized by a distinct pattern of expressed circRNAs. Except from an overall dissimilar profile of expressed circRNAs, the noticeably most overexpressed circRNAs seemed to differ between the four cell lines as well. Through the evaluation of these results, there seemed to be only a few circRNAs that were highly expressed in more than one cell line.

Based on the notion that higher intensity of the bands in agarose gels signifies higher concentration of the respective PCR product, various products were selected for Sanger sequencing. Through the annotation of the sequencing results, nine novel circRNAs deriving from the *PRMT1* gene were identified, named circ-PRMT1-1 to circ-PRMT1-9 (Figure 2). These circRNAs comprise two to six known exons of the *PRMT1* gene, while no intronic regions were found in their sequences. Their sequence length varies from 204 base pairs (circ-PRMT1-6) to 551 base pairs (circ-PRMT1-5). The most frequently represented exon is exon 6, which is found in eight out of nine circRNAs. In addition, these circRNAs had a distinct expression pattern (Table 1). More specifically, the first five circRNAs were found to be expressed in only one of the four cell lines, namely MDA-MB-468, circ-PRMT1-7 was found to be expressed in both the BT-20 and MCF-7 cell lines, while the remaining 3 *PRMT1* circRNAs were identified by reactions using the pool of the four cDNAs and were not detected in the individual reactions of each cell line. Interestingly, no specific association can be observed between the molecular subtypes of the cell lines (according to the PAM50 classification) and the expression pattern of the identified circRNAs. Additionally, no association in the expression pattern of the identified circRNAs was found between triple-negative (BT-20, MDA-MB-231, and MDA-MB-468) and non-triple-negative (MCF-7) cell lines.

### 3.2. Novel Splicing Events and Truncated Exons of the Novel PRMT1 CircRNAs

The back-splice junction of these novel circRNA molecules is of particular interest, since in each of the novel RNAs, the exons that form the back-splice junction were truncated at the respective end of each exon that participated in the junction (Table 1). In fact, these splice sites can be described as non-canonical, considering that the donor and acceptor sites are not characterized by the GT and AG bases that are commonly encountered in most splicing events. These alternative splice sites were observed in all of the novel circRNAs that were identified, and one or both exons that form the back-splice junction can be truncated. Intriguingly, the same exon may be differentially truncated, based on the circRNA in which it is found. These smaller exons have not been previously identified nor have they been found to participate in *PRMT1* linear transcripts.

### 3.3. Sequence Similarity in the Back-Splice Junction of Six Novel PRMT1 CircRNAs

Another interesting feature of the back-splice junction that emerged was that the truncated ends of the exons that form the junction share a short sequence similarity (Figure 3). In particular, there were several bases that can be aligned equally well in both truncated ends of the exons that form the back-splice junction with no errors and/or gaps. These identical sequences spanned 3–8 bases of the back-splice junction, and they were located within the exonic sequence and not at the end of the known *PRMT1* exons. Given that there was no parameter that could distinguish the exon to which these few bases belong, this short sequence at the final circRNA structure cannot be conclusively designated to a particular exon. This trait was observed in six out of the nine novel circRNAs, and it cannot be attributed to a particular cell line or specific *PRMT1* exons.

### 3.4. Predicted Interactions of the Novel PRMT1 CircRNAs and Pathway Analysis

Following the identification of novel circRNAs produced from *PRMT1*, extensive bioinformatics analysis was conducted to elucidate their putative interactions. Firstly, their miRNA-sponging ability was investigated. Indeed, all nine circRNAs are predicted to act as competing endogenous RNAs and sequester miRNAs, thus preventing them from regulating gene expression. The miRNAs that are predicted to bind to each novel *PRMT1* circRNA with the two highest prediction scores are shown in Table 2, while the binding motifs of these miRNAs in the predicted local secondary structure of each circRNA are depicted in Appendix A. The extensive list of all miRNAs that resulted from the analysis is presented in Appendix A. Out of these, eleven miRNAs were found to be specifically expressed and/or to possess a functional role in breast cancer according to PubMed^®^. The rest of the miRNAs have not been properly investigated in breast cancer, according to current literature. There are some prevalent miRNAs that these circRNAs are predicted to sequester, given that seven out of nine *PRMT1* circRNAs can sponge miR-494-3p, and six out of nine circRNAs can sponge miR-6754-3p with a high probability score. This can be attributed to the sequence similarity between circRNAs that comprise the same *PRMT1* exons.

Secondly, the potential of the novel *PRMT1* circRNAs to sequester RBPs was investigated, and all circRNAs were found to possess protein binding sites for various RBPs. The RBPs with the most binding sites and a high probability score are presented in Table 3, considering that a circRNA ought to possess a minimum number of binding sites for it to effectively sequester an RBP and affect its function. The complete list of results from the RBPmap tool is included in Appendix A. The most frequent RBP that is predicted to bind to all nine *PRMT1* circRNAs is the FUS RNA-binding protein, which is highly implicated in the regulation of transcription. Moreover, RNA-binding motif protein 45 (RBM45) was found to possess the most binging sites on the circRNA sequences, followed by serine/arginine-rich splicing factor 9 (SRSF9) and heterogeneous nuclear ribonucleoprotein F (HNRNPF). All these proteins are implicated in cancer progression through DNA damage response and signaling pathways, and their interaction with *PRMT1* transcripts could be critical to their complex functions.

Through the utilization of CLIP-seq publicly available data, several of these interactions could be verified. In particular, data from the CLIPdb database were available for the RBPs SRSF9, PCBP2, HNRNPF, and HNRNPK. Among these, SRSF9 and PCBP2 possess binding sites in the sequence of *PRMT1*, according to eCLIP and PAR-CLIP experiments. Moreover, the interaction of SRSF9, PCBP2 and HNRNPK was verified by data from the ENCORI database. CLIP-seq data were also available in ENOCRI for the MBLN1 protein; however, there were no binding sites on *PRMT1* based on the available data.

Based on the aforementioned results, human Kyoto Encyclopedia of Genes and Genomes (KEGG) pathway analysis was performed, by including the most frequent miRNAs that are predicted to bind to the novel *PRMT1* circRNAs and the mRNAs these miRNAs are most likely to target. Based on this analysis, the proteins that are produced from mRNAs that are targeted by the miRNAs participate most frequently in metabolic pathways and pathways in cancer, such as the calcium signaling pathway, cAMP signaling pathway, and glucagon signaling pathway (Appendix A).

### 3.5. Deduced Secondary Structure of PRMT1 CircRNAs and Novel PRMT1 Protein Isoforms

In order to gain insight into the characteristics of these novel circular molecules, in silico analysis was conducted for the prediction of the optimal secondary structures of the nine circRNA sequences, and the results are shown in Figure 4. Differences in structure, such as the relative abundance of loops, may have a direct effect on the sponging ability of each circRNA, rendering it more or less accessible to target miRNAs or RBPs. For instance, the increased paired residues in the structure of circ-PRMT1-8 could explain the scarce predicted binding sites for RBPs.

Considering that various circRNAs are reported to be translated and encode functional proteins, the query for ORFs in their nucleotide sequences followed. According to the results, five circRNAs contain an ORF and can potentially be translated, and the amino acid sequences of the putative PRMT1 protein isoforms were deduced. The start codon was the same in all instances and it resided in exon 5, while the stop codon differed between them. Following the amino acid sequence alignment from Clustal Omega (1.2.4.), it was evident that the five predicted PRMT1 isoforms possess the methyltransferase motifs I (VLDVGSGTG) and post-I (VIGIE) (Figure 5). These motifs are mainly responsible for the methyltransferase function and the stabilization of substrate binding. Interestingly, all the major motifs of the main PRMT1 isoform are found in the deduced amino acid sequence of PRMT1 is.circ5, indicating that it could perform a similar function and work supplementary to the main isoform.

Next, the existence of m^6^A modifications in the sequences of these five circRNAs was queried since this modification is associated with the initiation of circRNA translation. In fact, Yang et al. reported that one or two m^6^A sites are sufficient to promote circRNA translation [40]. Indeed, all circRNAs that possess an ORF were predicted to possess m^6^A sites as well (Table 4), and the local secondary structure of these sites is depicted in Appendix A. Considering that they share all five major PRMT1 motifs, 3D structure models of the main isoform and PRMT1 is.circ5 were built by the I-TASSER server based on the deduced amino acid sequence, and were visualized with PyMOL (Figure 6). While PRMT1 is.circ5 lacks a major structure part in comparison with the main isoform and does not possess the distinct combination of β-sheets, it contains the major motifs in an almost identical secondary structure with the main PRMT1 isoform. This similarity in structure might denote a similar methyltransferase function, since the predicted isoform possesses the motifs responsible for binding and stabilizing S-adenosylmethionine. This notion is also supported by the fact that both isoforms have the same binging sites for S-Adenosylhomocysteine (SAH). Moreover, the dimerization finger that is important for the interaction between PRMT1 and its substrate is also present in the predicted structure.

## 4. Discussion

The PRMTs constitute a group of proteins that play a key role in post-translational modifications. For example, cell cycle, signal transduction, RNA metabolism, chromatin structure, and RNA and protein trafficking can all be regulated by methylation [41]. PRMT1 is the most important enzyme in this family and methylates a wide range of molecules, including histone H4, heterogeneous nuclear ribonucleoproteins, and interleukin enhancer-binding factor protein 3 [42,43,44,45]. Changes in methylation patterns and histone abnormalities are key epigenetic modifications in cancer [46]. However, most attention so far has been given to the role of PRMT1 isoforms, and the transcriptional potential of *PRMT1* has not been sufficiently investigated [41].

CircRNAs deriving from this gene could be implicated in already-known *PRMT1* roles, either facilitating or obstructing them, or participate in pathways that have not yet been linked to *PRMT1* function. In this study, we investigated the possible role of *PRMT1* circRNAs in breast cancer cell lines. More specifically, the expression pattern of *PRMT1* circRNAs was examined in the MCF-7, MDA-MB-231, MDA-MB-468, and BT-20 cell lines, and we observed that a wide range of circRNAs is produced from *PRMT1*, highlighting the involvement of this gene in a variety of biological processes. However, the expression levels of most circRNAs deriving from this gene did not render them suitable for the current downstream analysis; most of these circRNAs could be identified by more high-throughput experiments, such as third-generation sequencing. According to the findings, apart from an overall distinct profile of expressed circRNAs, the most overexpressed circRNAs seem to differ between the four cell lines. This phenomenon may be attributed to the distinct characteristics of the cell lines which were used and, consequently, to a potentially strict cell line-specific expression pattern of these circRNAs. However, circ-PRMT1-7 was found in the BT-20 and MCF-7 cell lines, even though these cell lines differ in many aspects (histology, molecular subtype, tumor source). This fact may be due to reasons unrelated to such aspects and relevant to the general cellular environment, such as the availability of transcription factors and/or the expression levels of other molecules with regulatory roles, at the time of this circRNA transcription. Considering this, the clinical and pathological phenotypes of the four cell lines cannot be associated with a specific expression pattern of *PRMT1* circRNAs.

Alternative splicing events were observed in all novel *PRMT1* circRNAs, and the splicing pattern of each circRNA could potentially affect its expression levels and determine its spatial structure and mechanism of action [47,48]. A growing body of research suggests that aberrant alternative splicing is linked to the development of many malignancies [49,50,51,52]. As a result, investigating alternative splicing mechanisms adds to our knowledge of post-transcriptional regulatory patterns. Modified splicing patterns can be used as markers of a cellular state linked with cancer, even if they are not part of the fundamental disease process. Even though *PRMT1* expression has been examined in a variety of human tissues, the relative incidence of alternatively spliced *PRMT1* linear and circular transcripts varies across malignant breast tissues. Regarding the cell lines used in this study, all four alternative linear transcripts of *PRMT1* were found to be expressed, with transcript variant 1 (NM_001536.6) being the most prevalent [7,53]. Data from the present research evidence the existence of novel splicing events and truncated exons in the sequence of the novel *PRMT1* circRNAs, broadening our understanding of alternative splicing events of this gene, in the context of breast cancer. It is possible that *PRMT1* circRNA transcripts are preferentially produced in distinct subtypes of breast cancer, while aberrant alternative splicing of *PRMT1* circRNAs may act as a driver for breast cancer tumorigenesis through the deregulation of important molecular functions.

Another intriguing aspect of the back-splice junction that was discovered is that the shortened ends of the exons that make up the junction have a short sequence similarity; therefore, this brief sequence cannot be definitively assigned to a certain exon of the back-splice junction. This finding is rather puzzling and, according to our knowledge, this is the first study where this feature of the back-splice junction is reported. The sequence similarity in the exon ends that are back-spliced could indicate a novel step of circRNA biogenesis that has not been explored so far, since the currently known biogenesis mechanisms do not support this finding. For instance, an RBP could specifically recognize these highly similar sequences in the two exons that are going to be back-spliced, and by binding to them, it brings the exon ends to proximity. By the end of this interaction, the identified sequence by the RBP is featured once in the final circRNA structure.

Following the identification of novel circRNAs produced by *PRMT1*, a bioinformatics investigation was carried out to determine their interactions and their modes of action. The miRNA-sponging ability was firstly examined since it is the most well-described function of circRNAs [54]. For example, circ-ABCB10 is highly upregulated in breast carcinoma, and the repression of circ-ABCB10 decreased tumorigenesis while enhancing apoptosis. It was reported that circ-ABCB10 increases breast tumor progression by sponging miR-1271 [55]. According to our results, the novel *PRMT1* circRNAs could act as miRNA sponges. In particular, seven *PRMT1* circRNAs can sponge miR-494-3p with a high probability score, which is proved to regulate the renewal of breast cancer stem cells, as well as to affect the survival of breast cancer patients [56]. In addition, six *PRMT1* circRNAs can sponge miR-6754-3p, and this miRNA can bind to the 3’ UTR of several zinc finger proteins, such as ZNF99, ZNF486, and ZNF676. Therefore, its sequestering by the novel *PRMT1* circRNAs can actively affect its role in regulating transcription. Interestingly, ZNF486 was reported to reliably predict the prognosis of breast cancer patients, rendering it a promising candidate for further investigation as a *PRMT1* circRNAs target [57]. During KEGG pathway analysis, the downstream molecules whose expression might be affected by the miRNA-sponging function of *PRMT1* circRNAs are proteins that are mainly involved in metabolic pathways, such as the calcium metabolic pathway. Notably, PRMT1 was recently described as essential and responsible for the Ca^2+^-mediated erythroid differentiation [58]; therefore, circular variants of this gene might participate in the intracellular mediation of calcium signaling as well.

In addition, all novel circRNAs were found to possess protein binding sites for various RBPs. RBPs are a family of proteins that play a role in the metabolic processing of RNAs by influencing their maturation, transport, localization, and translation [59]. Recent research studies have revealed that the tertiary structure of RNA molecules has a significant impact on RNA–RBP interactions [60]. As a result, the novel circRNAs unique tertiary structure may influence their protein-binding capacity and it may also dictate whether a *PRMT1* circRNA would sponge RBPs rather than miRNAs. This is an interesting notion that has not been investigated thus far, considering that there are no reported circRNAs of the same gene with different modes of action, even though there are no indications to discourage this idea. According to the results of the bioinformatics analysis, RBM45, SRSF9, and HNRNPF were very likely to be sponged by *PRMT1* circRNAs, while FUS is predicted to bind to all of the novel circRNAs. These proteins are directly associated with cancer progression and are reported to interact as well; for instance, RBM45 acts competitively to histone deacetylase 1 (HDAC1) for binding to FUS as a DNA damage response, while SRSF9 and HNRPF are implicated in signaling and EMT regulation, since they modulate tumorigenesis through the WNT signaling pathway and EMT through SNAIL1 stabilization, respectively [61,62,63]. Moreover, the binding of FUS could be another means for *PRMT1* to exert its role in modulating genes expression, in conjunction with the function of its linear transcripts.

Last but not least, several novel *PRMT1* circRNAs also possess an ORF and predicted m^6^A sites in their sequence. These findings suggest that the *PRMT1* circRNAs could potentially be translated, and the m^6^A modification is proved to facilitate circRNA translation without the need for an internal ribosome entry site (IRES). Since it is a reversible epitranscriptomic modification, it could drive the translation of circRNAs which normally possess an additional function, such as sponging, under specific conditions such as cellular stress, deregulated proliferation, and genome instability. It is widely known that alternative splicing events, such as exon skipping and the formation of novel exon boundaries, are prevalent in cancer, and alternative isoforms are associated with the regulation of the hallmarks of cancer. Interestingly, the predicted is.circ5 contains all three PRMT1 methyltransferase domains, the S-adenosylmethionine-binding motif and the dimerization finger that interacts with the outer surface of this binding site, and they have almost the same spatial conformation as in the main isoform. Therefore, the active site and topological domains are retained in the predicted isoform and, as a result, it could work synergistically with PRMT1, and dimmers from partly different PRMT1 isoforms might be formed between them [6].

It is evident that the regulatory pathways in which *PRMT1* circular transcripts are involved, impacting breast cancer development and progression, are elaborate and intertwining, rendering the function of this gene even more complex than previously thought. It would be worthwhile to examine the expression and functions of the novel circRNAs in both normal and cancerous conditions, in order to draw more definitive conclusions on their interactions and regulatory potential. Further studies focusing on their quantification and expression levels in breast cancer patients’ serum or other bodily fluids would better solidify their potential as reliable breast cancer biomarkers. Another interesting approach would be to examine their expression in exosomes, given that exosomal secretion of various circRNAs is reported to affect breast cancer aggressiveness, mostly through aberrant signaling [64], as well as therapy resistance [65]. Moreover, the possible functions of these circRNAs could be validated by visualizing their distribution and cellular localization by circFISH [66], or by verifying in which compartments are the circRNAs localized by circRNA expression profiling among the subcellular fractions [67]. In addition, the circRNA–miRNA interactions could be further examined by tagging a novel *PRMT1* circRNA for affinity pulldown assays using a biotinylated antisense oligonucleotide that targets the back-splice junction and can be pulled down specifically with streptavidin-coated beads [68]. However, there lies a limitation of our study. In order to confirm that a circRNA is successfully enriched in the pulldown, qPCR analysis is needed, and due to the similarity in sequence between the novel circRNAs, it would be difficult to ensure that only the desired circRNA is amplified. There are still various unexplored aspects of circRNA biology, and as research efforts continue towards their elucidation, circRNAs will acquire better established roles and therapeutic potential.

## 5. Conclusions

Overall, circRNA biogenesis, expression profiles, unique functional roles, and physiological importance have received a great deal of attention recently, and many intriguing aspects of circRNAs are continuously emerging. As a result, circRNAs may outperform traditional protein-based cancer indicators, which are frequently imprecise. Moreover, breast cancer transcriptome analyses are essential for the identification of risk genes; however, integrating data from breast cancer subtype-specific studies has been limited. Further transcriptomic analyses focused on circRNAs are expected to provide a more comprehensive picture of individual patients’ tumors. In particular, circRNAs deriving from genes that are already established risk factors in breast cancer, such as *PRMT1*, constitute highly promising targets for transcriptomics studies. However, we are still making the important first steps on this subject and several unknown factors need to be investigated and elucidated for circRNAs, with emphasis on their functional roles and the manner in which they regulate major cellular processes.

## Figures and Tables

**Figure 1 genes-13-01133-f001:**
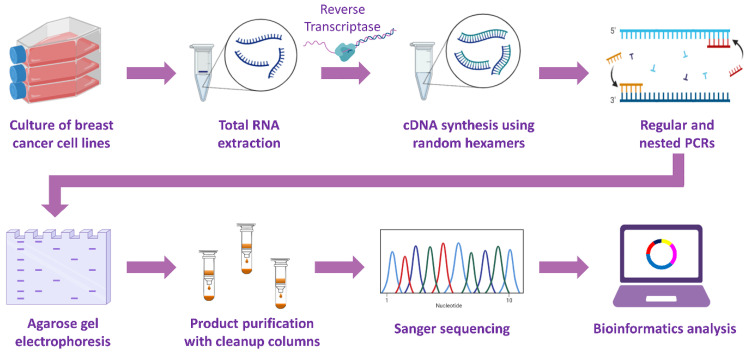
Overview of the experimental workflow that was followed in this research article for the identification of novel *PRMT1* circRNAs in breast cancer cell lines.

**Figure 2 genes-13-01133-f002:**
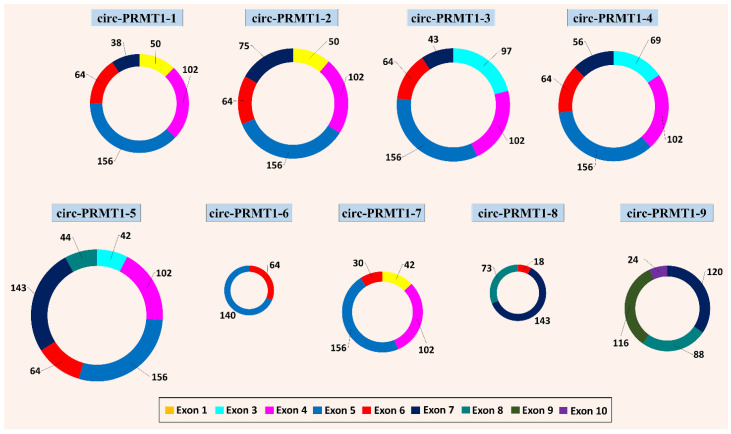
The 9 novel *PRMT1* circRNAs that were identified, depicted in scale based on their size. The exon composition of each circRNA is shown, according to the color-coding that is shown at the bottom of the figure. The length of each exon (in nucleotides) is also depicted at the end of the respective black lines.

**Figure 3 genes-13-01133-f003:**
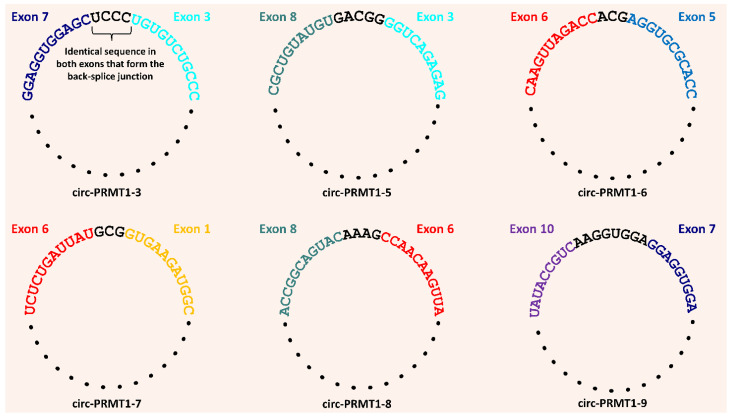
Depiction of the short sequence similarity that was observed in the exon ends that form the back-splice junction in 6 novel *PRMT1* circRNAs. These identical sequences that belong at both exons of the back-splice junction are shown in black, while the rest of the exonic sequence is shown in each exon’s distinctive color. Black dots signify the rest of the circRNA sequence that is not portrayed for comprehensibility reasons. The names of the circRNAs in which this feature was observed are found below each circRNA depiction.

**Figure 4 genes-13-01133-f004:**
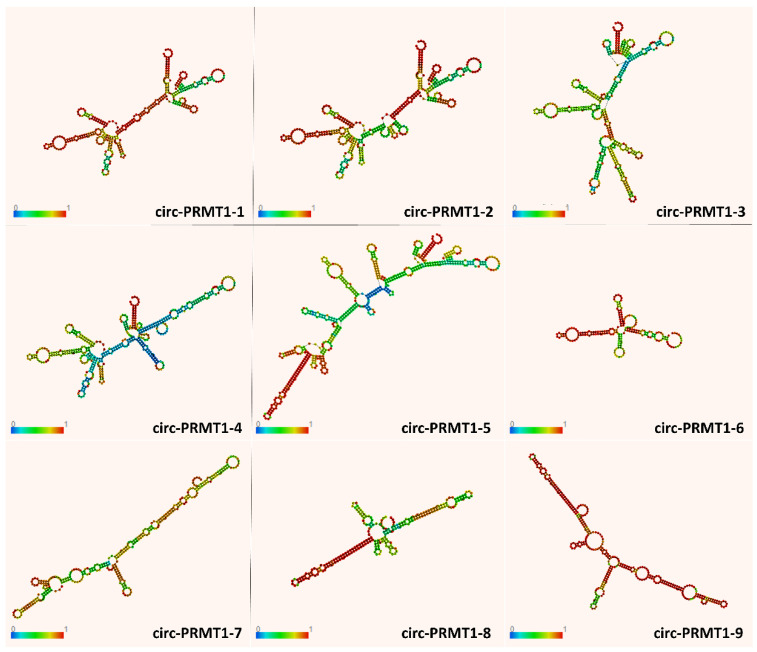
Graphical visualization of the secondary structures and base-pair probabilities of the novel *PRMT1* circRNAs, using a loop-based energy model and dynamic programming algorithm of the RNAfold prediction tool [36]. The color scale denotes the accuracy of the minimum free energy and partition function: dark blue signifies the lowest base-pairing probability, while dark red signifies the highest base-pairing probability.

**Figure 5 genes-13-01133-f005:**
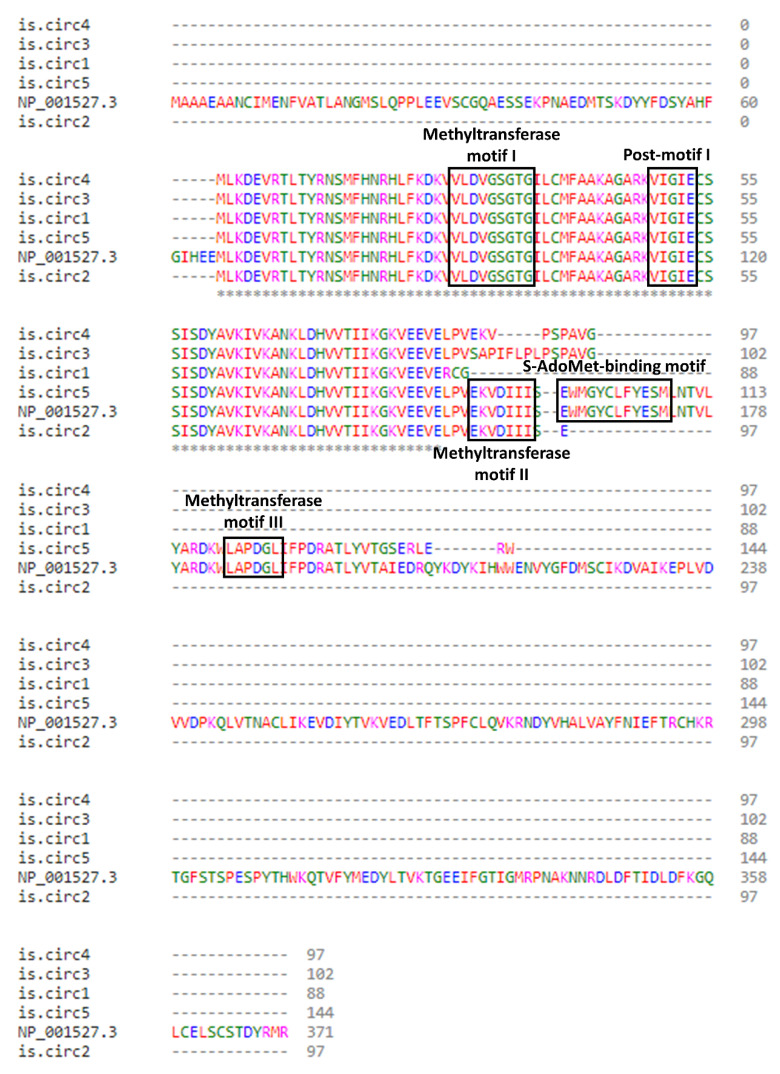
Clustal omega multiple sequence alignment of the amino acid sequences derived from the translation of the 5 *PRMT1* circRNAs that possess an open reading frame, including the sequence of the main PRMT1 isoform (NP_001527.3). All PRMT1 isoforms are predicted to possess the signature methyltransferase motif I and Post-I, while two of them possess the methyltransferase motif II, and PRMT1 is.circ5 additionally possess the methyltransferase motif III and the S-adenosylmethionine (AdoMet)-binding motif (black boxes). An * (asterisk) indicates positions which have a single identical residue. The color-coding represents the physicochemical properties of residues: red = small, hydrophobic, including aromatic minus Y; blue = acidic; magenta = basic; green = hydroxyl, sulfhydryl, amine, plus G.

**Figure 6 genes-13-01133-f006:**
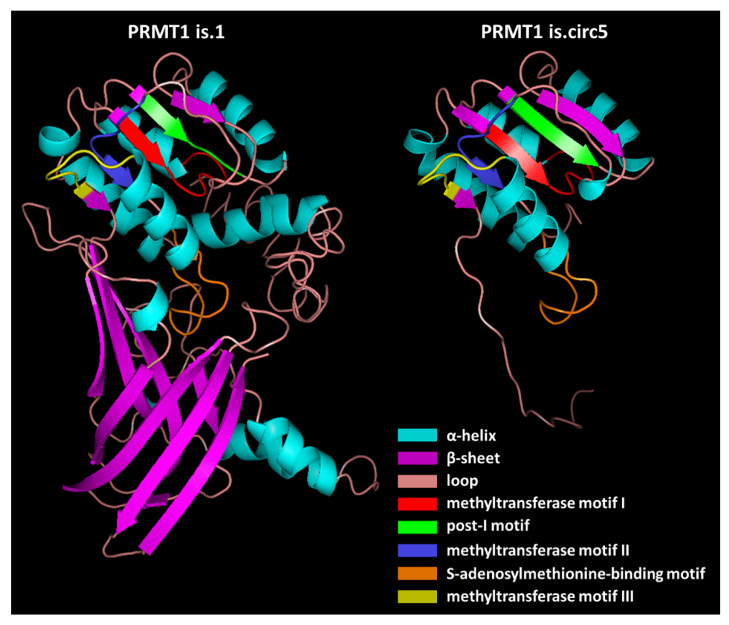
Predicted models of the main PRMT1 isoform (NP_001527.3) and the novel PRMT1 isoform that is potentially produced from circ-PRMT1-5, visualized with PyMOL. The I-TASSER server was used for the prediction, and only the 3D structure with the highest confidence score is displayed for both cases. The color code for the secondary structures and the shared main domains between the isoforms is also displayed.

**Table 1 genes-13-01133-t001:** Features of the 9 novel *PRMT1* circRNAs that were identified and their GenBank^®^ accession numbers. For each circRNA, the cell line cDNA(s) in which it was found expressed is shown, as well as how many and which *PRMT1* exons they comprise.

*PRMT1* CircRNA	Accession Number	Cell Line CDNA(s)	Exon Count	Exons Included in the CircRNA Structure ^1^	Length (nt ^2^)
circ-PRMT1-1	ON081037	MDA-MB-468	5	Exon 1, Exon 4, Exon 5, Exon 6, Exon 7 *	410
circ-PRMT1-2	ON081038	MDA-MB-468	5	Exon 1, Exon 4, Exon 5, Exon 6, Exon 7 *	447
circ-PRMT1-3	ON081039	MDA-MB-468	5	Exon 3 *, Exon 4, Exon 5, Exon 6, Exon 7 *	462
circ-PRMT1-4	ON081040	MDA-MB-468	5	Exon 3 *, Exon 4, Exon 5, Exon 6, Exon 7 *	447
circ-PRMT1-5	ON081041	MDA-MB-468	6	Exon 3 *, Exon 4, Exon 5, Exon 6, Exon 7, Exon 8 *	551
circ-PRMT1-6	ON081042	Mix of BT-20, MCF-7, MDA-MB-231, MDA-MB-468	2	Exon 6, Exon 5 *	204
circ-PRMT1-7	ON081043	BT-20, MCF-7	4	Exon 1 *, Exon 4, Exon 5, Exon 6 *	330
circ-PRMT1-8	ON081044	Mix of BT-20, MCF-7, MDA-MB-231, MDA-MB-468	3	Exon 6 *, Exon 7, Exon 8 *	234
circ-PRMT1-9	ON081045	Mix of BT-20, MCF-7, MDA-MB-231, MDA-MB-468	4	Exon 7 *, Exon 8, Exon 9, Exon 10 *	348

^1^ The *PRMT1* exons that are truncated and form the back-splice junction are denoted with an asterisk (*). The truncation site of an exon can vary, depending on the circular transcript. ^2^ nucleotides.

**Table 2 genes-13-01133-t002:** The miRNAs that are predicted to bind to the novel *PRMT1* circRNAs with the two highest prediction scores, on a scale up to 100. For each miRNA, their binding motif on the respective circRNA sequence is shown as well.

*PRMT1* CircRNA	Target MiRNA	Prediction Score	Binding Motif
circ-PRMT1-1	miR-6754-3p	80	GGUGAAGA
	miR-494-3p	76	AUGUUUCA
circ-PRMT1-2	miR-6754-3p	77	GGUGAAGA
	miR-494-3p	76	AUGUUUCA
circ-PRMT1-3	miR-494-3p	76	AUGUUUCA
	miR-6754-3p	68	GGUGAAGA
circ-PRMT1-4	miR-494-3p	76	AUGUUUCA
	miR-6754-3p	68	GGUGAAGA
circ-PRMT1-5	miR-494-3p	75	AUGUUUCA
	miR-6754-3p	67	GGUGAAGA
circ-PRMT1-6	miR-494-3p	78	AUGUUUCA
	miR-6754-3p	73	GGUGAAGA
circ-PRMT1-7	miR-494-3p	76	AUGUUUCA
	miR-1306-3p	66	GGAGGUG
circ-PRMT1-8	miR-4745-3p	65	CCGGGCCA
	miR-1538-3p	62	CCGGGCCA
circ-PRMT1-9	miR-4696-5p	86	GUCUUGCA
	miR-588-5p	71	GUGGCCA

**Table 3 genes-13-01133-t003:** The RNA-binding proteins (RBPs) that are predicted to bind to the novel *PRMT1* circRNAs, based on the number of binding sites and a high probability score, as resulting from the RBPmap tool.

*PRMT1* CircRNA	RBP	Number of Binding Sites	*p*-Value ^1^
circ-PRMT1-1	RNA-binding motif protein 45 (RBM45)	24	1.25 × 10^−2^
	Serine/arginine-rich splicing factor 9 (SRSF9)	14	9.37 × 10^−3^
	Heterogeneous nuclear ribonucleoprotein F (HNRNPF)	14	1.57 × 10^−2^
circ-PRMT1-2	RNA-binding motif protein 45 (RBM45)	24	1.25 × 10^−2^
	Serine/arginine-rich splicing factor 2 (SRSF2)	16	9.57 × 10^−3^
	Heterogeneous nuclear ribonucleoprotein F (HNRNPF)	15	1.52 × 10^−2^
circ-PRMT1-3	RNA-binding motif protein 45 (RBM45)	25	1.46 × 10^−2^
	Poly(rC) binding protein 2 (PCBP2)	24	7.57 × 10^−3^
	RNA-binding motif protein 23 (RBM23)	16	6.94 × 10^−3^
	RNA-binding motif protein 24 (RBM24)	16	1.82 × 10^−2^
circ-PRMT1-4	RNA-binding motif protein 45 (RBM45)	23	1.28 × 10^−2^
	Heterogeneous nuclear ribonucleoprotein F (HNRNPF)	15	1.36 × 10^−2^
	Serine/arginine-rich splicing factor 9 (SRSF9)	14	1.28 × 10^−2^
circ-PRMT1-5	RNA-binding motif protein 45 (RBM45)	31	1.53 × 10^−2^
	Heterogeneous nuclear ribonucleoprotein F (HNRNPF)	17	1.43 × 10^−2^
	Serine/arginine-rich splicing factor 9 (SRSF9)	16	1.33 × 10^−2^
	RNA-binding motif protein 24 (RBM24)	15	1.26 × 10^−2^
circ-PRMT1-6	Muscleblind-like splicing regulator 1 (MBNL1)	7	2.73 × 10^−2^
circ-PRMT1-7	RNA-binding motif protein 45 (RBM45)	29	9.24 × 10^−3^
	RNA-binding motif protein 41 (RBM41)	12	1.25 × 10^−2^
circ-PRMT1-8	Serine/arginine-rich splicing factor 9 (SRSF9)	12	4.42 × 10^−3^
circ-PRMT1-9	RNA-binding motif protein 45 (RBM45)	23	1.52 × 10^−2^
	Heterogeneous nuclear ribonucleoprotein K (HNRNPK)	12	3.11 × 10^−2^

^1^ The *p*-value refers to the probability of obtaining a significant match (statistically significant: *p* < 0.050).

**Table 4 genes-13-01133-t004:** The predicted m^6^A sites of the novel *PRMT1* circRNAs that possess an open reading frame (ORF), according to the SRAMP tool. For each m^6^A site, the exact position on the respective circRNA sequence and the overall score is presented. The overall prediction score is calculated by combining the scores from K-nearest neighbor encoding, positional binary encoding, and nucleotide pair spectrum encoding. The specific adenine (A) which is modified, as well as its secondary structure, are shown in bold.

*PRMT1* CircRNA	Position of m^6^A Site(s)	Sequence Context	Local Secondary Structure	Score
circ-PRMT1-1	355	GAGG**A**CAUG	PP**P**PP ^1^	0.59
circ-PRMT1-2	392	GAGG**A**CAUG	PP**P**PP	0.59
circ-PRMT1-3	334	AGAG**A**CUGG	PI**P**PP ^2^	0.61
	407	GAGG**A**CAUG	PP**M**MM ^3^	0.60
circ-PRMT1-4	319	AGAG**A**CUGG	PI**P**PP	0.62
	392	GAGG**A**CAUG	PP**M**MM	0.60
circ-PRMT1-5	423	AGAG**A**CUGG	PI**P**PP	0.61
	496	GAGG**A**CAUG	PP**M**MM	0.60

^1^ P, paired residues; ^2^ I, interior loop; ^3^ M, multiple loop.

## Data Availability

Not applicable.

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
