# Peer review of "Identification of Novel Circular RNAs of the Human Protein Arginine Methyltransferase 1 (PRMT1) Gene, Expressed in Breast Cancer Cells"

_genes, 2022, doi:10.3390/genes13071133_

Round 1

Reviewer 1 Report

The manuscript titled “Identification of novel circular RNAs of the human protein arginine methyltransferase 1 (PRMT1) gene, expressed in breast cancer cells” describes the authors found 9 new circRNAs from 4 breast cancer cell lines. The authors further use bioinformatics methods to predict the possible binding sites and pathways of these circRNAs and prospect their potential targets for cancer therapy. The followings are some concerns and comments have been pointed out that the authors may want to consider.

1. Lines 28-29 Keywords: The keyword “back-splicing” appears 3 times in the main context; “histone methylation” and “miRNA sponging” did not appear in the main context; only one time that “transcriptomics” appears in the main context; I’d suggest the authors consider some other more suitable keywords.

2. Line 31 introduction section: A total of 5/23 (22%) self-citation in the introduction part seems too much. I’d highly suggest the authors revise this part. PLEASE DO NOT simply by adding more references to dilute self-citation.

3.  Line 46: Please add references to the first sentence.

4.  Line 50: Please add the reference for “10 genes…”

5.   Lines 56-60: Please add references to the first two sentences.

6.  Lines 62-65: Please add references to “Moreover,…breast cancer cells”.

7. Line 94 Materials and Methods section: Please provide details information for all the reagents and instruments to make your work reproducible relatively easier.

8.     Line 106: Please provide more details and protocol.

9.     Line 173: Please use italic p as it refers to a p-value. Check throughout the manuscript.

10. Line 237 Table 1: Did the authors have any ideas that a) why is circ-PRMT1-7 only found in BT-20 and MCF-7? b) Why does MDA-MB-468 includes all circRNAs except circ-PRMT1-7. I’d highly suggest the authors add more description, and/or additional discussion if necessary.

11.  Line 317: Please provide the methods used to obtain p-values.

12.  Line 320 Figure 4: Please provide the tool for RNAfold prediction.

13.  Line 371 Figure 6: Please provide the tool for 3D structures.

14. I’d suggest the authors a) include the description that why chose MCF-7, MDA-MB-231, MDA-MB-468, and BT-20 cell lines; b) are there any possible differences between TNBC cell line and none TNBC cell line;

15. I’d suggest the author provide a summary of the mechanism figure if you do not mind.

16.  Table S2: Please present primers as pairs.

Author Response

REVIEWER’S Comments and Corresponding Responses

Reviewer #1 (Comments to the Author):

 1.       Lines 28-29 Keywords: The keyword “back-splicing” appears 3 times in the main context; “histone methylation” and “miRNA sponging” did not appear in the main context; only one time that “transcriptomics” appears in the main context; I’d suggest the authors consider some other more suitable keywords.

We thank the Reviewer for this suggestion. We decided to replace the keywords “back-splicing”, “histone methylation”, “miRNA sponging” and “transcriptomics” with the keywords “back-splice junction”, “miRNAs”, “bioinformatics” and “RNA modification”. The new keywords appear several times in the main context.

2.       Line 31 introduction section: A total of 5/23 (22%) self-citation in the introduction part seems too much. I’d highly suggest the authors revise this part. PLEASE DO NOT simply by adding more references to dilute self-citation.

According to the Reviewer’s comment, we revised this part of the Introduction, and 3/5 self-citations were removed. The other citations are essential for the Introduction, considering that Scorilas et al. previously described the genomic organization and expression pattern of the PRMT1 gene. In place of the removed references, the following relevant papers were cited: 20.      Kristensen, L.S.; Jakobsen, T.; Hager, H.; Kjems, J. The emerging roles of circRNAs in cancer and oncology. Nat Rev Clin Oncol 2022, 19, 188-206, doi:10.1038/s41571-021-00585-y.21.      Huang, M.S.; Zhu, T.; Li, L.; Xie, P.; Li, X.; Zhou, H.H.; Liu, Z.Q. LncRNAs and CircRNAs from the same gene: Masterpieces of RNA splicing. Cancer Lett 2018, 415, 49-57, doi:10.1016/j.canlet.2017.11.034.26.      Arnaiz, E.; Sole, C.; Manterola, L.; Iparraguirre, L.; Otaegui, D.; Lawrie, C.H. CircRNAs and cancer: Biomarkers and master regulators. Semin Cancer Biol 2019, 58, 90-99, doi:10.1016/j.semcancer.2018.12.002.

3.       Line 46: Please add references to the first sentence.

We have added the following reference for this sentence:8.        Baldwin, R.M.; Morettin, A.; Cote, J. Role of PRMTs in cancer: Could minor isoforms be leaving a mark? World J Biol Chem 2014, 5, 115-129, doi:10.4331/wjbc.v5.i2.115.

4.       Line 50: Please add the reference for “10 genes…”

We have added the following reference at the end of this period:10.       Walsh, T.; King, M.C. Ten genes for inherited breast cancer. Cancer cell 2007, 11, 103-105, doi:10.1016/j.ccr.2007.01.010.

5.       Lines 56-60: Please add references to the first two sentences.

We have added the following references to this part:13.       Suresh, S.; Huard, S.; Brisson, A.; Nemati, F.; Dakroub, R.; Poulard, C.; Ye, M.; Martel, E.; Reyes, C.; Silvestre, D.C.; et al. PRMT1 Regulates EGFR and Wnt Signaling Pathways and Is a Promising Target for Combinatorial Treatment of Breast Cancer. Cancers (Basel) 2022, 14, doi:10.3390/cancers14020306.14.       Nakai, K.; Xia, W.; Liao, H.W.; Saito, M.; Hung, M.C.; Yamaguchi, H. The role of PRMT1 in EGFR methylation and signaling in MDA-MB-468 triple-negative breast cancer cells. Breast cancer 2018, 25, 74-80, doi:10.1007/s12282-017-0790-z.

6.       Lines 62-65: Please add references to “Moreover,…breast cancer cells”.

We have added the following reference for this sentence:15.      Gao, Y.; Zhao, Y.; Zhang, J.; Lu, Y.; Liu, X.; Geng, P.; Huang, B.; Zhang, Y.; Lu, J. The dual function of PRMT1 in modulating epithelial-mesenchymal transition and cellular senescence in breast cancer cells through regulation of ZEB1. Sci Rep 2016, 6, 19874, doi:10.1038/srep19874.

7.       Line 94 Materials and Methods section: Please provide details information for all the reagents and instruments to make your work reproducible relatively easier.

We thank the Reviewer for this comment. We have added details and clarifications in order for the Materials and Methods section to be adequately descriptive. Thus, the following sentences were added to make the experimental workflow easier to reproduce: ·         Page 3, lines 107-108: […] at a final concentration of 0.01 mg/mL.·         Page 3, lines 135-136: The process was performed in a MiniAmp Thermal Cycler (Applied Biosystems™, Thermo Fisher Scientific Inc.), …·         Page 3, line 147: […] at a final volume of 100μL. ·         Page 4, line 148: […] using 1μL of the diluted products as template …·         Page 4, line 152: […] KAPA Taq Buffer A, in which MgCl2 at a final concentration of 1.5 mM was included, …·         Page 4, lines 159-160: […] electrophoresed on 2% agarose gels in equal volumes (10μL) …·         Page 4, lines 163-167: […] as per the manufacturer’s instructions. Briefly, for each 100 mg of agarose gel, 200μL Buffer NTI was used to solubilize the gel slice. Next, the samples were centrifuged for 30 sec at 11,000 g, and two steps of wash with Buffer NT3 and centrifugation were performed. At the final step, 30μL of Buffer NE were added and each sample was centrifuged for 1 min at 11,000 g in order to elute the PCR product. Moreover, the Ta for each primer pair for the annealing step of the first- and second-round PCRs is shown in Supplementary Table S2, for replication purposes.

8.       Line 106: Please provide more details and protocol.

As suggested by the Reviewer, we added the following details to this part:

  • Page 3, lines 117-121: More specifically, the absorbance at 260nm was measured for the assessment of RNA concentration, and the absorbance at 280nm and 230nm were used for the estimation of protein and phenol contamination. All A260/A280 and A260/A230 ratios were within the acceptable values.
  • Page 3, lines 123-133: […] and random hexamer primers (New England Biolabs Ltd., Hitchin, UK), which is a mixture of single-stranded random hexanucleotides with 5'- and 3'-hydroxyl ends. In this way, the cDNA produced from each sample was derived not only from mRNA molecules but from circRNAs as well. Each reaction mixture contained, besides the RNA volume, 1 μL of random hexamer primers, 1 μL 10 mM dNTP Mix (10 mM of each dATP, dGTP, dCTP and dTTP at neutral pH) and DEPC- treated H2O at a final volume of 10 μL. The mixture was heated at 65°C for 5 min, and was quickly transferred on ice for 1 min. Next, in each RNA sample, 1 μL of 5X First-Strand Buffer, 0.1 M DTT, 40 units RNaseOUT™ recombinant RNase inhibitor (Invitrogen™, Thermo Fisher Scientific Inc., Waltham, MA, USA) and 200 units of Reverse Transcriptase were added, resulting in a final reaction volume of 20 μL. Finally, each reaction was inactivated at 70°C for 15 minutes.

9.       Line 173: Please use italic p as it refers to a p-value. Check throughout the manuscript. 

We thank the Reviewer for this comment. We have corrected this issue throughout the manuscript.

 10.   Line 237 Table 1: Did the authors have any ideas that a) why is circ-PRMT1-7 only found in BT-20 and MCF-7? b) Why does MDA-MB-468 includes all circRNAs except circ-PRMT1-7. I’d highly suggest the authors add more description, and/or additional discussion if necessary. 

  1. When assessing the characteristics of the 9 novel PRMT1 circRNAs, circ-PRMT1-7 does not seem to process a specific attribute that distinguishes it from the rest. The fact that it was found in BT-20 and MCF-7, even though these cell lines differ in many aspects (histology, molecular subtype, tumor source) may be due to reasons unrelated to such aspects and relevant to the general cellular environment, such as the availability of transcription factors and/or the expression levels of other molecules and enzymes with regulatory roles at the time of this circRNAs’ transcription. Considering this, the clinical and pathological phenotypes of the 4 cell lines cannot be associated with a specific expression pattern of PRMT1 circRNAs
  2. We cannot be certain that MDA-MB-468 includes all circRNAs except circ-PRMT1-7, considering that 3 PRMT1 circRNAs were identified by reactions using the pool of the 4 cDNAs and were not detected in the individual reactions of each cell line. Therefore, it is possible that these circRNAs are not expressed in all 4 cell lines.

According to the Reviewer’s suggestion, we added the following text to the Discussion:

  • Page 14, lines 453-458: However, circ-PRMT1-7 was found in the BT-20 and MCF-7 cell lines […] a specific expression pattern of PRMT1 circRNAs.

11.   Line 317: Please provide the methods used to obtain p-values.

Prompted by the Reviewer’s suggestion, we added a summary of the methods used to calculate the p-values by the RBPmap tool in the Materials and Methods section:

  • Page 5, lines 195-199: The Z-score measures the deviation of the site's weighted rank (WR) score from the mean score calculated using the genome-specific background. The Z-score is coupled to a p-value which represents the probability of obtaining a specific Z-score considering a normal one-tailed distribution. The RBPmap tool requires that the final WR score of a site will be significantly greater (p-value<0.05) than the mean score calculated for the background, in order to consider this site as a predicted binding site.

12.   Line 320 Figure 4: Please provide the tool for RNAfold prediction. 

We have added the respective reference of the RNAfold prediction tool in the legend of Figure 4:

35.       Lorenz, R.; Bernhart, S.H.; Honer Zu Siederdissen, C.; Tafer, H.; Flamm, C.; Stadler, P.F.; Hofacker, I.L. ViennaRNA Package 2.0. Algorithms Mol Biol 2011, 6, 26, doi:10.1186/1748-7188-6-26.

13.   Line 371 Figure 6: Please provide the tool for 3D structures.

 We have added the respective reference of the PyMOL tool in the legend of Figure 6:

39.       Lilkova, E., et al. The PyMOL Molecular Graphics System, Schrodinger, LLC.: 2015.

14.   I’d suggest the authors a) include the description that why chose MCF-7, MDA-MB-231, MDA-MB-468, and BT-20 cell lines; b) are there any possible differences between TNBC cell line and none TNBC cell line;

We thank the reviewer for the insightful comments. Prompted by this, we have adjusted the following sections of the manuscript: ·         Page 3, lines 100-104: “Four elaborately characterized and well-studied, in the context of transcriptomic analysis, breast cancer cell lines were chosen in this study. The BT-20, MCF-7, MDA-MB-231, and MDA-MB-468 breast cancer cell lines which are of distinct characteristics and molecular subtypes (Table S1) were purchased from the American Type Culture Collection (ATCC®) and cultured in an incubator at 37 °C and adjusted CO2 concentration of 5%.”·         Page 7, lines 271-275: “Interestingly, no specific association can be observed between the molecular subtypes of the cell lines (according to the PAM50 classification) and the expression pattern of the identified circRNAs. Additionally, no association in the expression pattern of the identified circRNAs was found between triple-negative (BT-20, MDA-B-231, and MDA-MB-468) and non-triple-negative (MCF-7) cell lines.

15.   I’d suggest the author provide a summary of the mechanism figure if you do not mind.

According to the Reviewer’s suggestion, we revised the relevant part in the Results and the Discussion, in order for it to be clearer and more comprehensible. More specifically, this section of the Results was divided into distinct paragraphs. In addition, Figure 3 does not depict a mechanism, it is an illustration of our observations regarding the intriguing aspects of the back-splice junction of the novel circRNAs. Moreover, the Discussion of this part was altered and improved in terms of clarity, and any mention of a mechanism throughout this section refers to an already known biogenesis mechanism. The respective parts of the manuscript that changed are the following: ·     Page 8, lines 294-303: Another interesting feature of the back-splice junction that emerged was that the truncated ends of the exons that form the junction share a short sequence similarity (Figure 3). In particular, there were several bases that can be aligned equally well in both truncated ends of the exons that form the back-splice junction with no errors and/or gaps. These identical sequences spanned 3-8 bases of the back-splice junction, and they were located within the exonic sequence and not at the end of the known PRMT1 exons. Given that there was no parameter that could distinguish the exon to which these few bases belong, this short sequence at the final circRNA structure cannot be conclusively designated to a particular exon. This trait was observed in 6 out of the 9 novel circRNAs, and it cannot be attributed to a particular cell line or specific PRMT1 exons.·     Page 15, lines 479-489: Another intriguing aspect of the back-splice junction that was discovered is that the shortened ends of the exons that make up the junction have a short sequence similarity; therefore, this brief sequence cannot be definitively assigned to a certain exon of the back-splice junction. This finding is rather puzzling, and according to our knowledge, this is the first study where this feature of the back-splice junction is reported. The sequence similarity in the exon ends that are back-spliced could indicate a novel step of circRNA biogenesis that has not been explored so far, since the currently known biogenesis mechanisms do not support this finding. For instance, an RBP that is not yet associated with circRNA biogenesis could specifically recognize these identical highly similar sequences in the two exons that are going to be back-spliced, and by binding to them, it brings the exon ends to proximity. By the end of this interaction, the identified sequence by the RBP is featured once in the final circRNA structure.

16.   Table S2: Please present primers as pairs.

As suggested by the Reviewer, we modified Supplementary Table S2 accordingly. Moreover, the Ta for each primer pair for the annealing step of the first- and second-round PCRs was added in Table S2, for replication purposes.

The authors wish to thank the Reviewer for the constructive comments that led to the improvement of the current manuscript.

Reviewer 2 Report

circRNAs have been shown to act as transcriptional regulators, microRNA sponges and protein template. In this manuscript, Papatsirou et al. identified 9 novel circRNAs from PRMT1 in 4 breast cancer cell lines using PCR and sanger sequencing technologies. The authors also discussed their potential functions of miRNA/protein sponges. While the authors have made an amount of effort, there are some points that need to be adequately addressed.

1) Did authors use RNase R to treat total RNAs before reverse transcription?  Is there any enrichment process that was performed?

2) How is the abundance of these circRNAs in breast cancer cell lines?

3) Are there any different circRNAs from other genes in breast cancers?

4)  In the Table2/TableS3, how many miRNAs were specifically expressed in breast cancer?

5)  Where are the miRNAs target positions on predicted secondary structures of these circRNAs? The authors should prepare a similar figure like FigS2.

6) Without any experiment verification, it is hard to convince readers to believe that miRNA sponges functions of these novel circRNAs. The authors should design at least one experiment to perform the verification.

7) There are many published RBPs CLIP sequencing data. The authors should also use these public databases to verify the interaction between RBPs and circRNAs.

 In summary, the authors identified novel circRNAs in breast cancers. This finding is interesting and important. However, additional functional assays are necessary.

Author Response

Reviewer #2 (Comments to the Author):

1.       Did authors use RNase R to treat total RNAs before reverse transcription?  Is there any enrichment process that was performed?

During the experimental pipeline of this work, the RNAs were not treated with RNase R prior to reverse transcription. In order to ensure that circRNAs were reversely transcribed, we used random hexamer primers instead of an oligo-dT–adaptor primer that is typically used. Moreover, with the use of divergent primers for the first- and second-round PCR assays we ascertained that no linear transcripts could be amplified, and only products deriving from circRNAs were obtained. This information is incorporated into the text in the following parts:

  • Page 3, lines 122-126: Then, reverse transcription was carried out, using 3 μg of each RNA sample and random hexamer primers (New England Biolabs Ltd., Hitchin, UK), which is a mixture of single-stranded random hexanucleotides with 5'- and 3'-hydroxyl ends. In this way, the cDNA produced from each sample was derived not only from mRNA molecules but from circRNAs as well.
  • Page 3, lines 141-144: To selectively amplify cDNAs that originate from circRNAs of the PRMT1 gene, and not its linear RNA variants, divergent primers were designed (Table S2). These primers are facing outward, in contrast to the conventionally used convergent primers, and they are specific for exons of the PRMT1 gene.

 2.       How is the abundance of these circRNAs in breast cancer cell lines? 

In the course of the qualitative expression analysis that was performed in the breast cancer cell lines, the expression pattern of circRNAs differed between the 4 cell lines. The 9 novel circRNAs that were identified in this study were chosen for Sanger sequencing due to their increased expression based on the bands’ intensity on the agarose gels. Additionally, there were several other products of each cell lines’ cDNA, indicating a plethora of circRNAs deriving from PRMT1 in these cell lines. However, the expression levels of these circRNAs did not render them suitable for the current downstream analysis; these circRNAs could be identified by more high-throughput experiments, such as 3rd generation sequencing.  Accordingly, the following remarks were added to the text:

  • Page 15, lines 443-449: More specifically, the expression pattern of PRMT1 circRNAs was examined, […] such as 3rd generation sequencing.

 3.       Are there any different circRNAs from other genes in breast cancers?

 Numerous circRNAs from different genes have been identified in breast cancer over the past few years. Some indicative examples are mentioned in the revised Introduction section:·         Page 2, lines 83-86: Moreover, over the past few years, numerous circRNAs from different genes have been identified in breast cancer, the proteins of which are important regulators in different cellular processes. Some indicative examples are circ-HER2/HER2-103, circCD44, and circACTN4.  Additionally, the following relevant references were added to the text:27.       Li, J.; Ma, M.; Yang, X.; Zhang, M.; Luo, J.; Zhou, H.; Huang, N.; Xiao, F.; Lai, B.; Lv, W.; et al. Circular HER2 RNA positive triple negative breast cancer is sensitive to Pertuzumab. Mol Cancer 2020, 19, 142, doi:10.1186/s12943-020-01259-6.28.       Li, J.; Gao, X.; Zhang, Z.; Lai, Y.; Lin, X.; Lin, B.; Ma, M.; Liang, X.; Li, X.; Lv, W.; et al. CircCD44 plays oncogenic roles in triple-negative breast cancer by modulating the miR-502-5p/KRAS and IGF2BP2/Myc axes. Mol Cancer 2021, 20, 138, doi:10.1186/s12943-021-01444-1.29.       Wang, X.; Xing, L.; Yang, R.; Chen, H.; Wang, M.; Jiang, R.; Zhang, L.; Chen, J. The circACTN4 interacts with FUBP1 to promote tumorigenesis and progression of breast cancer by regulating the expression of proto-oncogene MYC. Mol Cancer 2021, 20, 91, doi:10.1186/s12943-021-01383-x.

4.       In the Table2/TableS3, how many miRNAs were specifically expressed in breast cancer?

Eleven miRNAs were found to be specifically expressed and/ or to possess a functional role in breast cancer according to PubMed®. The rest of the miRNAs have not been properly investigated in breast cancer, according to the current literature. This information was incorporated into the text:·         Pages 8-9, lines 320-323: Out of these, […] according to current literature. 

5.       Where are the miRNAs target positions on predicted secondary structures of these circRNAs? The authors should prepare a similar figure like FigS2.

 Prompted by the Reviewers’ suggestion, we prepared a new Figure (Figure S2) that depicts the miRNAs target positions on the predicted secondary structures of the novel PRMT1 circRNAs. Figure S2 is referenced in the manuscript as such:·         Page 8, lines 316-319: The miRNAs that are predicted to bind to each novel PRMT1 circRNA with the two highest prediction scores are shown in Table 2, while the binding motifs of these miRNAs in the predicted local secondary structure of each circRNA are depicted in Figure S2. The new Figure S2 legend is the following one: Figure S2: Local secondary structures of the predicted miRNAs target positions in the novel PRMT1 circRNAs. The local secondary structures were predicted using the RNAfold tool, and the binding motifs of the miRNAs resulted from miRDB. The bases that constitute each miRNAs binding motif are outlined in red color.

6.       Without any experiment verification, it is hard to convince readers to believe that miRNA sponges functions of these novel circRNAs. The authors should design at least one experiment to perform the verification.

We agree that the regulatory function of the novel PRMT1 circRNAs needs further elucidation in order to have robust evidence of their role. However, functional analysis of 9 novel circRNAs that include several same exons, therefore, share sequence similarity, and are found only through pre-amplification so far is rather challenging. Moreover, functional experiments are also very time-consuming and costly, rendering such an attempt not feasible in the framework of the current study. Such interactions could be identified by tagging a PRMT1 circRNA for affinity pulldown using a biotinylated ASO that targets the back-splice junction and can be pulled down specifically with Streptavidin-coated beads. However, in order to confirm that the circRNA is successfully enriched in the pulldown, qPCR analysis is needed and due to the similarity in sequence, it would be difficult to ensure that only the desired circRNA is amplified. Prompted by the Reviewers comment, the Discussion was revised accordingly in order to address these limitations of our study: ·                     Page 16, lines 556-565 Moreover, the possible functions of these circRNAs could be validated by visualizing their distribution and cellular localization by circFISH [65], or by verifying in which compartments are the circRNAs localized by circRNA expression profiling among the subcellular fractions [66]. In addition, the circRNA-miRNA interactions could be fur-ther examined by tagging a novel PRMT1 circRNA for affinity pulldown assays using a biotinylated antisense oligonucleotide that targets the back-splice junction and can be pulled down specifically with streptavidin-coated beads [67]. However, there lies a limitation of our study. In order to confirm that the circRNA is successfully enriched in the pulldown, qPCR analysis is needed, and due to the similarity in sequence between the novel circRNAs, it would be difficult to ensure that only the desired circRNA is amplified.

7.       There are many published RBPs CLIP sequencing data. The authors should also use these public databases to verify the interaction between RBPs and circRNAs.

We thank the Reviewer for this comment. According to this suggestion, we utilized two public databases, CLIPdb and ENCORI, and searched for CLIP sequencing data of the RBPs that are included in Table 3. Considering that all 9 novel PRMT1 circRNAs are firstly identified in this study, there is no CLIP sequencing data that verify the interaction between there circRNAs and the RBPs that are predicted to bind to them. Alternatively, we searched for RBP-mRNA interactions, by selecting the PRMT1 mRNA, as an indirect way of verifying the binding of these RBPs to PRMT1 sequences. The Methods and Results were revised to include this information:

  • Page 5, lines 202-210: As a means of further verifying […] by selecting the RBP of interest and PRMT1 as the target.
  • Page 9, lines 341-347: Through the utilization of CLIP-seq publicly available data, […] binding sites on PRMT1 based on the available data.

The authors wish to thank the Reviewer for the constructive comments that led to the improvement of the current manuscript.

Round 2

Reviewer 1 Report

The manuscript has been well improved. I only have the following minor comments that the authors should consider and double-check to homogenous the format throughout the manuscript again before publication.  Good luck.

1) Lines 122-123: “All A260/A280 and A260/A230 ratios were within the acceptable values.” is not a clear and scientific description.

2) Please be consistent with or without a space between the value and the unit throughout the manuscript.

3) Line 279: The “MDA-B-231” should be MDA-MB-231”.

Reviewer 2 Report

The manuscript is much improved after the revision.